# PRETRAIN-TO-FINETUNE ADVERSARIAL TRAINING VIA SAMPLE-WISE RANDOMIZED SMOOTHING

## ABSTRACT

Developing certified models that can provably defense adversarial perturbations is important in machine learning security. Recently, randomized smoothing, combined with other techniques (Cohen et al., 2019; Salman et al., 2019), has been shown to be an effective method to certify models under $l_2$ perturbations. Existing work for certifying $l_2$ perturbations added the same level of Gaussian noise to each sample. The noise level determines the trade-off between the test accuracy and the average certified robust radius. We propose to further improve the defense via sample-wise randomized smoothing, which assigns different noise levels to different samples. Specifically, we propose a pretrain-to-finetune framework that first pretrains a model and then adjusts the noise levels for higher performance based on the model's outputs. For certification, we carefully allocate specific robust regions for each test sample. We perform extensive experiments on CIFAR-10 and MNIST datasets and the experimental results demonstrate that our method can achieve better accuracy-robustness trade-off in the transductive setting.

## 1 INTRODUCTION

The vulnerability of neural networks to adversarial examples has attracted considerable attention in safety-critical scenarios. For example, adding visually indistinguishable perturbations to input images would misguide a deep classifier to make wrong predictions (Szegedy et al., 2013; Goodfellow et al., 2014). Such an intriguing property of neural networks has spawned a lot of works on training robust neural networks and certifying network robustness with theoretical guarantees.

Many heuristic defense algorithms have been developed (Szegedy et al., 2013; Goodfellow et al., 2014; Moosavi-Dezfooli et al., 2016; Papernot et al., 2016; Kurakin et al., 2016; Carlini & Wagner, 2017; Athalye et al., 2018), together with many empirically robust models intended to defend certain types of adversarial perturbations. However, there is no theoretical guarantees of robustness for many heuristic defense algorithms and many of them have been broken subsequently by more carefully designed and more powerful attack algorithms (Athalye et al., 2018). This motivates the development of certifiably robust classifiers whose outputs are guaranteed to be the same within a $l_p$-ball of certain radius, hence can defend any adversarial pertubation smaller than that radius (Hein & Andriushchenko, 2017; Raghunathan et al., 2018; Wong & Kolter, 2017; Gowal et al., 2018; Wong et al., 2018; Zhang et al., 2018; Cohen et al., 2019; Salman et al., 2019).

Recently, Cohen et al. (2019) utilized a randomized smoothing technique to build robust smoothed classifiers with provable $l_2$-robustness. Specifically, the smoothed classifier outputs the class most likely to be predicted by the base classifier when the input is perturbed by a certain level of Gaussian noise. Building upon this idea, Salman et al. (2019) combined randomized smoothing with adversarial training and achieved a state-of-the-art provable l2-defense. Both Cohen et al. (2019) and Salman et al. (2019) added the same level of Gaussian noise to each sample, and a higher noise level leads to lower accuracy but a larger average robust radius. While it is well known that accuracy and robustness are at odds with each other (Tsipras et al., 2018), we find that we can improve both accuracy and robustness via sample-wise randomized smoothing, which assigns different noise levels to different samples. While the above argument seems intuitive, we note that there is a subtle and challenging issue: a close examination of the proof of the randomized smoothing theorem in Cohen et al. (2019) dictates that we cannot assign arbitrary noise level to any test point; a certain robustness radius around a point can be certified only if all points within the radius are assigned the same Gaussian variance. To

address this issue, we divide the input space into "robust regions", and samples in the same region are assigned the same noise level. To certify a sample-wise randomized smoothed model, we make sure that the certified $l_2$-ball is entirely contained by one of the regions, so that the randomized smoothing theorem can be applied to that region. Hence, precisely speaking, in our method, the classification of a testing point depends on which "robust regions" it lies in, and hence on other testing points in the test dataset. Hence, we present our results in the transductive setting (the test dataset is known), but our method also works in the setting where the test data points come online in an i.i.d. fashion (see the detailed discussion in Appendix D).

Our contributions can be summarized as follows:

1. We introduce a pretrain-to-finetune framework that first pretrains a model with random noise levels and then further finetunes it with specific selected noise levels, which maximizes the robust radius of each train set sample based on the pretrained model's output.

2. In transductive setting, we allocate specific robust regions and assign noise levels for test set data using linear programming to get near optimal results. If the test data points come online, we allocate robust regions for them one by one. If the newly test sample falls in an allocated region, then it uses the same noise level. Otherwise, we allocate a brand new region for it.

3. We conduct a series of experiments on CIFAR-10 and MNIST to evaluate the performance of our method. Comparing with the state-of-the-art algorithms, Smooth-Adv (Salman et al., 2019) and Macer (Zhai et al., 2020), our sample-wise method achieves $40\%$ improvements on average certified radius on CIFAR-10 and comparable results on MNIST.

## 2 RELATED WORK

There are a large body of works proposing different adversarial defense algorithms, which can be divided into two categories, empirical defenses and provable defenses, by whether the resulting model achieves certified robustness.

**Empirical defenses** Empirical defenses train classifiers that are robust under specific attacks. Lacking theoretical guarantees, these defenses can usually be broken by stronger attacks. Early defenses added noise to the input sample or features in inference in hopes that the adversarial perturbation would degenerate into random noise which could be easily handled. For example, Papernot et al. (2016) used a distillation framework to remove the effect of adversarial examples on the model. Guo et al. (2017) preprocessed the input images before feeding them into the networks, such as JPEG compression. However, Athalye et al. (2018) found that these empirical defenses could be broken by stronger attacks.

So far, adversarial training (Goodfellow et al., 2014; Madry et al., 2017), the process of training a model over on-the-fly adversarial examples, has been one of the most powerful empirical defenses. Later directly minimize the classification errors of adversarial examples generated by project gradient descend (PGD) which maximize the errors, actually solving a min-max optimization problem. The TRADES technique (Zhang et al., 2019) improved the results by minimizing a surrogate loss, consisting of the natural error and boundary error, and provided a tight upper bound to the original loss. Though powerful, adversarial training still lacks a certified guarantee.

**Provable defenses** Due to the fact that empirical defenses are only robust to some specific attacks and face potential threats, the focus of the research switches to the certified defenses whose predictions can be proved robust when perturbed within a certain range. Gradually more certified defenses have been proposed recently.Raghunathan et al. (2018) certified a shallow network based on semidefinite relaxation, Wong & Kolter (2017) made use of convex outer adversarial polytope based on relu activation function and Zhang et al. (2018) proposed a solution for general activation function. Though effective, they all rely on the structure of the network.

A randomized smoothing classifier (Cohen et al., 2019) is a certified robust classifier independent of the structure of the network and only relies on the prediction of the base classifier. In fact, it is a virtual classifier and its prediction is generated from the prediction of the base classifier. Then Salman et al. (2019) proposed Smooth-Adv which can directly attack the virtual classifier and promoted

performance with a large margin. Based on these, we propose sample-wise randomized smoothing to improve certified robustness further.

## 3 PRELIMINARIES

### 3.1 NOTATION

We study the task of image classification. Every image is represented by $x \in \mathcal{X}$ and $\mathcal{X}$ stands for the image space $[0,1]^{h \times w \times c}$, where $h, w, c$ represent the height, width and depth of the picture respectively. For a classification task of $k$ classes, we intend to build a classifier $f : \mathcal{X} \rightarrow \mathcal{Y} = \{0, 1, ..., k-1\}$. For a neural network, $f_\theta(x)$ represents the prediction of the sample $x$ and variable $\theta$ represents the parameters of the neural network. $f_\theta$ predicts the class that achieves the maximum value at the last layer of the network.

### 3.2 ADVERSARIAL EXAMPLES

Given a classifier $f_\theta$ and an input-label pair $(x, y)$, we say that $x'$ is an adversarial example to $x$ if $\mathcal{D}(x, x') < \epsilon$, $f_\theta(x) = y$ and $f_\theta(x') \neq y$. The attack algorithm seeks for any different prediction result in the $\epsilon$-ball centered at $x$ (i.e., $\{x' \mid \mathcal{D}(x, x') < \epsilon\}$). $\mathcal{D}(\cdot, \cdot)$ can represent different distance metric such as $l_2$ and $l_\infty$ metric. We only focus on $l_2$ distance in this paper and we use $\mathcal{D}(\cdot, \cdot)$ to denote the $l_2$ distance. For empirical defenses, researchers have built heuristic robust classifiers so that specific attack algorithms can hardly find adversarial examples $x'$ around $x$. For certified defenses, they build certified robust classifiers $g_\theta$ and look for the maximum value of $\epsilon$ with which for any perturbation $\delta$ such that $\mathcal{D}(x + \delta, x) < \epsilon$, one can rigorously prove that $g_\theta(x + \delta) = g_\theta(x)$. Such $\epsilon$ is also called the *robustness radius* about $x$.

### 3.3 RANDOMIZED SMOOTHING

The randomized smoothing method (Cohen et al., 2019) is a powerful defense framework and can be applied to any classification model. Suppose $f_\theta$ is the base model. The smoothed model $g_\theta(x, \sigma)$ predicts the category $c$ most likely to be predicted by $f_\theta$ when the input is sampled from a Gaussian distribution $\mathcal{N}(x, \sigma^2 I)$, where $\sigma$ is a carefully chosen standard deviation. Formally, $g_\theta$ is defined as follows:

$$g_\theta(x, \sigma) = \arg\max_{c \in \mathcal{Y}} P(f_\theta(x + \eta) = c) \qquad \text{where} \quad \eta \sim \mathcal{N}(0, \sigma^2 I) \tag{1}$$

Cohen et al. (2019) proved that $g_\theta$ is provably robust to adversarial perturbations under the $l_2$ norm constraint. Define $c_A = \arg\max_c P(f_\theta(x + \eta) = c)$, $p_A = P(f_\theta(x + \eta) = c_A)$ and $p_B = \max_{c \neq c_A} P(f_\theta(x + \eta) = c)$. We can prove that $g_\theta$ always predicts $c_A$ in $\{z \in \mathcal{X} : \mathcal{D}(z, x) < r\}$, where the robust radius $r$ is given by the following formula:

$$r = R(f_\theta, \sigma, x) = \frac{\sigma}{2}(\Phi^{-1}(p_A) - \Phi^{-1}(p_B)) \tag{2}$$

where $\Phi^{-1}$ is the inverse of the standard Gaussian CDF. This theorem (called the randomized smoothing theorem) implies that $g_\theta$ is provably $l_2$-robust at $x$ if it predicts the label of $x$ correctly.

To train provably robust models, Cohen et al. (2019) proposed to train and certify the model with the same noise level, exactly the same standard deviation. Furthermore, (Salman et al., 2019) equipped randomized smoothing with adversarial training, and proposed an attack algorithm, called Smooth-Adv. By randomly sampling multiple Gaussian noise around each sample during the training procedure, the algorithm directly estimates $g_\theta$ and seeks for adversarial examples for $g_\theta$. The attack algorithm is shown in Eq. (3) and $\mathcal{L}_{CE}$ represents the cross entropy loss.

$$x' = \arg\max_{\mathcal{D}(x', x) < \epsilon} \mathcal{L}_{CE}(g_\theta(x', \sigma), y)$$

$$= \arg\max_{\mathcal{D}(x', x) < \epsilon} \left( -\log\left(\frac{1}{m}\sum_{i=1}^{m} f_\theta(x' + \eta_i)_y\right)\right) \tag{3}$$

Though heuristic for adversarial training, combining with randomized smoothing and directly optimizing $g_\theta$ has greatly improve the certifiable robustness. Based on Smooth-Adv, we apply sample-wise randomized smoothing to further improve the defense.

# 4 SAMPLE-WISE RANDOMIZED SMOOTHING

## 4.1 MOTIVATION

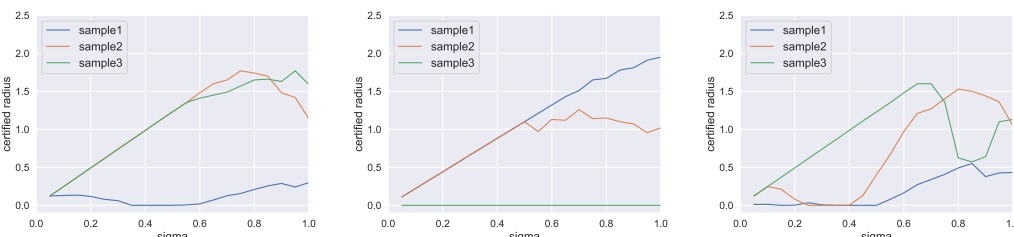

(a) Smooth-adv with test samples  (b) Pretrain model with train samples (c) Finetune model with test samples

Figure 1: Certified radius of several samples with different Gaussian distribution noise under three models. (a) Smooth-adv model test with test set samples. (b) Our pretrain model test with train set samples. (c) Our finetune model test with test set samples which are the same as (a).

From Eq. (1) and Eq. (2) we can see that the prediction of the model and the robust radius it achieves at each sample are highly correlated to the level of added Gaussian noise. The choice of $\sigma$ has a great impact on the model's accuracy as well as robustness. Previous works built the smoothed model by adding the same level of noise to the entire dataset, which is not optimal for each individual sample. If we add a large noise to a sample near the decision boundary, it is very likely for the sample to be misclassified. On the other hand, for samples far away from the decision boundary, adding a large noise allows us to achieve a larger certified radius. Ideally, we need to use small $\sigma$ for samples near the decision boundary, and big $\sigma$ for samples far away.

To provide a motivating example, we randomly select several test samples from the CIFAR-10 test set and certify them with different $\sigma$ under one pretrained Smooth-Adv model provided by Salman et al. (2019) which was trained with $\sigma = 1.0$. We take the base classifier out and build upon it different smoothed classifiers with $\sigma$ from 0 to 1.0 with 0.05 as the interval. Then we certify the smoothed classifier with Eq. (2) and plot the result in Fig. 1(a). We can see that the optimal $\sigma$ is not 1.0. By assigning the optimal $\sigma$ to each individual sample, we can achieve larger certified radii than reported in the original paper.

## 4.2 SAMPLE-WISE RANDOMIZED SMOOTHING

The above example motivates us to design a sample-wise randomized smoothing algorithm which assigns different noise levels to different samples. In this section, we elaborate the details of this framework. Generally speaking, the training procedure consists of two stages: the pretrain stage learns the data distribution with randomly assigned $\sigma$, and the finetune stage improves the performance of the model by selecting the optimal $\sigma$.

**Pretrain-to-Finetune Framework**     In order to assign the most suitable $\sigma$ to every sample, we first train a base classifier $f_\theta$, and then select the optimal $\sigma$ for each sample based on this classifier. At the first pretraining stage, we utilize the Smooth-Adv as shown in Eq. (3) to train the base classifier $f_\theta$ with uniform random $\sigma_i^j$ applied to $j_{th}$ sample in $i_{th}$ epoch as described in Eq. (4).

$$\sigma_i^j \sim \text{Uniform}(\sigma_{min}, \sigma_{max}) \tag{4}$$

Having pretrained the base classifier $f_\theta$, we certify every sample in the train set with various $\sigma$ using Eq. (5). $h : \mathcal{X} \times \mathcal{Y} \to \{0, 1\}$ indicates whether $g_\theta$ predicts right or wrong with given $\sigma$. When $g_\theta(x^j, \sigma) = y^j$, $h$ takes the value 1 otherwise 0. From the Eq. (5), we would like to find a $\sigma$ that maximizes the certified radius under the premise of correct prediction. The basic approach is to use a

grid search, with $\sigma_{interval}$ being the size of the grid:

$$\sigma_{train}^j = \arg\max_\sigma \left( R(f_\theta, x^j, \sigma) \times h_{g_\theta, \sigma}(x^j, y^j) \right)$$

$$\text{where} \quad \sigma \in \{\sigma = t \times \sigma_{interval} \mid \sigma_{min} \leq \sigma \leq \sigma_{max}, t \in \mathbb{N}\}$$

(5)

---

**Algorithm 1** Training of our proposed framework

---

**Input:** Training dataset $S$. Batchsize $B$. Noise range $(\sigma_{min}, \sigma_{max})$ and interval $\sigma_{interval}$. Epochs $T_{pretrain}$ and $T_{finetune}$. Learning rate $lr_{pretrain}$ and $lr_{finetune}$. Number of noise sample $m$ and attack steps $k$.

1: ATTACKER ← Smooth-Adv
2: **while** epoch $i < T_{pretrain}$ **do**
3:     Generate $\sigma_i^j = Uniform(\sigma_{min}, \sigma_{max})$ for $1 \leq j \leq len(S)$
4:     **while** iterations $t \leq len(S)/B$ **do**
5:         Sample batch images $x^{k_1}, x^{k_2}, ..., x^{k_B}, 1 \leq k_1, k_2, ..., k_B \leq len(S)$
6:         Generate noise samples $\eta_q^{k_p} \sim \mathcal{N}(0, \sigma_i^{k_p})$ for $1 \leq p \leq B, 1 \leq q \leq m$
7:         Generate adversarial examples $x_q^{k_p}$ for $x^{k_p}$ using k-steps ATTACKER with $\eta_q^{k_p}$
8:         Calculate $\mathcal{L}_{CE}$ by $x_q^{k_p} + \eta_q^{k_p}$ and update $\theta$ with $lr_{pretrain}$
9:     **end while**
10: **end while**
11: Generate specified $\sigma^j = \arg\max_\sigma \left( R(f_\theta, x^j, \sigma) \times h_{g_\theta, \sigma}(x^j, y^j) \right)$ for $1 \leq j \leq len(S)$
12: **while** epoch $i < T_{finetune}$ **do**
13:     **while** iterations $t \leq len(S)/B$ **do**
14:         Sample batch images $x^{k_1}, x^{k_2}, ..., x^{k_B}, 1 \leq k_1, k_2, ..., k_B \leq len(S)$
15:         Generate noise samples $\eta_q^{k_p} \sim \mathcal{N}(0, \sigma^{k_p})$ for $1 \leq p \leq B, 1 \leq q \leq m$
16:         Generate adversarial examples $x_q^{k_p}$ for $x^{k_p}$ using k-steps ATTACKER with $\eta_q^{k_p}$
17:         Calculate $\mathcal{L}_{CE}$ by $x_q^{k_p} + \eta_q^{k_p}$ and update $\theta$ with $lr_{finetune}$
18:     **end while**
19: **end while**

---

Fig. 1(b) shows the certified radius of the pretrained model on several samples in train set under various noise levels. We can see that the model has different behavior on different samples. Sample 1 is far away from the decision boundary, so its certified radius keeps growing as $\sigma$ increases. Sample 2 is closer to the decision boundary and achieves its optimal $\sigma$ at 0.7. Sample 3 is mis-classified by the base classifier, and its certified radius is always 0.

After we find the optimal $\sigma_{train}^j$ for each training sample, we further finetune the model $f_\theta$ with the optimal noise levels assigned at the second stage, thereby improving the performance of the smoothed classifier. The whole framework for training is shown in Algorithm 1. Fig. 1(c) shows the certified radius of the same samples in Fig. 1(a) with the finetuned model.

**Predicting Procedure**     At test time, we divide the input space $\mathcal{X}$ into a set of balls $\{B_i\}_{i \geq 1}$ (which we call robust regions) that do not intersect with each other. Within $B_i$ the standard deviation of the Gaussian noise is fixed at $\sigma_{B_i}$, so that we can apply the randomized smoothing theorem within each $B_i$. $x_{test}^j$ lies in the robust region $B_{i_j}$ and uses the standard deviation $\sigma_{test}^j = \sigma_{B_{i_j}}$. Define the neighborhood $B(x^j, r^j) = \{x : \mathcal{D}(x, x^j) < r^j\}$ where $r^j$ stands for the robust radius of $x^j$. We have to make sure that $B(x^j, r^j) \subset B_{i_j}$ or we cannot apply the randomized smoothing theorem. The next problem is how to divide the input space to make $r^j$ as large as possible.

We consider the case where the test samples arrive one by one (in the appendix we will consider the other case where test samples arrive in batches). As shown in Algorithm 2, when a new test example $x_{test}^{j'}$ arrives, we first check whether it falls in any allocated robust region $B_i \in C^{j'} = \{B_{i_j} : 1 \leq j < j'\}$. If $x_{test}^{j'} \in B_i$, then we simply let $i_{j'} = i$. The robust radius $r^{j'}$ in this case can be obtained according to Eq. (6), where $\partial B_{i_{j'}}$ stands for the boundary of the ball $B_{i_{j'}}$ and $d(\cdot, \cdot)$ is the Euclidean distance function.

$$r^{j'} = \min \left\{ R(f_\theta, x_{test}^{j'}, \sigma_{test}^{j'}), d(x_{test}^{j'}, \partial B_{i_{j'}}) \right\}$$

(6)

If the new test sample $x_{test}^{j'}$ does not belong to any allocated robust region $B_i$, then we will create a new one from the unallocated part of the input space $\mathcal{X}$, which is essentially a ball centered at $x_{test}^{j'}$. First we find the optimal noise level $\sigma_{test}^{j'}$ using Eq. (7). It is slightly different from Eq. (5) which we used during training because we have no access to the true label at test time.

$$\sigma_{test}^{j'} = \arg\max_{\sigma} \left( \mathrm{R}(f_\theta, x_{test}^{j'}, \sigma) \right) \tag{7}$$
$$\text{where} \quad \sigma \in \{\sigma = t \times \sigma_{interval} | \sigma_{min} \leq \sigma \leq \sigma_{max}, t \in \mathbb{N}\}$$

Then we define the new region $B_{i_{j'}}$ according to Eq. (8). It ensures that the new region would not intersect with any of the old ones.

$$r^{j'} = \min \left\{ \mathrm{R}(f_\theta, x_{test}^{j'}, \sigma_{test}^{j'}), \min_{B_i \in C^{j'}} d(x_{test}^{j'}, \partial B_i) \right\}, \quad B_{i_{j'}} = B(x_{test}^{j'}, r^{j'}) \tag{8}$$

---

**Algorithm 2** Allocation of Regions and Assignment of Standard Deviations

---

**Input:** Testing dataset S.
**Output:** $B_j, \sigma_{B_j}, \sigma_{test}^j, B(x^j, r^j), (1 \leq j \leq len(S))$
 1: **for** $j' = 1, \cdots, len(S)$ **do**
 2: $\quad C^{j'} = \{B_{i_j} : 1 \leq j < j'\}$
 3: $\quad$ **for each** $B_i$ **in** $C^{j'}$ **do**
 4: $\quad\quad$ **if** $x_{test}^{j'} \in B_i$ **then**
 5: $\quad\quad\quad i_{j'} = i$
 6:
 7: $\quad\quad\quad \sigma_{test}^{j'} = \sigma_{B_i}$
 8: $\quad\quad\quad r^{j'} = \min \left\{ \mathrm{R}(f_\theta, x_{test}^{j'}, \sigma_{test}^{j'}), d(x_{test}^{j'}, \partial B_i) \right\}$
 9: $\quad\quad\quad$ Continue to the next $j'$
10: $\quad\quad$ **end if**
11: $\quad$ **end for**
12: $\quad$ *// No $B_i$ found, create a new one*
13: $\quad \sigma_{test}^{j'} = \arg\max_{\sigma} \left( \mathrm{R}(f_\theta, x_{test}^{j'}, \sigma) \right)$
14: $\quad r^{j'} = \min \left\{ \mathrm{R}(f_\theta, x_{test}^{j'}, \sigma_{test}^{j'}), \min_{B_i \in C^{j'}} d(x_{test}^{j'}, \partial B_i) \right\}$
15: $\quad B_{i_{j'}} = B(x_{test}^{j'}, r^{j'})$
16: **end for**

---

Now we define our final model $G : \mathcal{X} \to \mathcal{Y}$ by Eq. (9) where we assign the noise level $\sigma_{test}^j$ to sample $x^j$.

$$G(x^j) = g_\theta(x^j, \sigma_{test}^j) = \arg\max_{c \in \mathcal{Y}} P(f_\theta(x^j + \eta) = c) \quad \text{where} \quad \eta \sim \mathcal{N}(0, \sigma_{test}^j{}^2 I) \tag{9}$$

For the sample $x^j$, since we do not know its label, the principle of choosing the $\sigma$ is to choose the $\sigma$ that can maximize the certified radius. Such a prediction method will choose an appropriate $\sigma$ according to test sample's location and the property, which selects the larger noise level to promote robustness or selects the smaller noise level to ensure accuracy, thus achieving better accuracy-robustness trade-off.

As the regions are allocated one by one, the order we allocate regions affects the performance of these provable robustness. In the transductive setting, we can find the near optimal solution by *linear programming* instead of allocating individually. The details of linear programming and how much the order affects the robustness are shown in Appendix C.

## 5 EXPERIMENTS

In this section, we evaluate our sample-wise randomized smoothing method on CIFAR-10 and MNIST and compare it with existing state-of-the-art models, Smooth-Adv(Salman et al., 2019) and

Macer(Zhai et al., 2020). We have conducted extensive experiments and the result can show the following results: (1) our method can improve the provable robustness significantly over existing methods on CIFAR-10. (2) our method achieves comparable results on MNIST. (3) Ablation studies show that our sample-wise randomized smoothing method should be applied in both training and testing to get better results (Appendix B.2). (4) we can achieve nearly optimal allocation of regions using linear programming in transductive setting (Appendix C).

## 5.1 SETTINGS

Following the experiment settings in (Salman et al., 2019; Zhai et al., 2020), we use the same base classifier ResNet-110 for CIFAR-10 and LeNet for MNIST. As our sample-wise method is trained based on Smooth-Adv, the hyper-parameters are the same as those in (Salman et al., 2019) and the details can be found in Appendix A. Specially, we replace the fixed standard deviation $\sigma$ by a range $[\sigma_{min}, \sigma_{max}]$ and it has three choices: $\{[0, 12, 0.25], [0.12, 0.50], [0.12, 1.00]\}$. $\sigma_{interval}$ is set to be a fixed value 0.05. We pretrain the models for 150 epochs and finetune them for another 150 epochs. For certification, we use only 500 noise samples to select specific $\sigma$ for train set and 100,000 noise samples for test set.

## 5.2 RESULTS

In this section, we first compare our sample-wise method with Smooth-Adv as our training process is based on Smooth-Adv. Then we compare with Smooth-adv and Macer on the metric *average certified radius*(ACR), suggested by (Zhai et al., 2020). The average certified radius is defined as $\frac{1}{n}\sum_{j=1}^{n} r^j \times h_{g_\theta, \sigma^j}(x^j, y^j)$, and $h$ is defined in Section 4.2.

Fig. 2 depicts the certified accuracy of two methods given the same hyper-parameters except $\sigma$. The models are trained with attack steps $k = 10$, maximum $l_2$ perturbation $\epsilon = 2.0$ and number of noise samples $m = 2$. In Fig. 2, **Smooth-Adv-0.25** stands for the Smooth-Adv model trained with $\sigma = 0.25$ and so on. Our sample-wise model use 3 different ranges of $\sigma$, for example the **Ours-[0.12-0.25]** stands for the model whose $\sigma$ ranges from 0.12 to 0.25.

Fig. 2 shows our sample-wise model performs better than the baseline model Smooth-Adv. With the larger noise level, the improvement is more pronounced. This is because with larger $\sigma$, the baseline model performs poorer. However, with more careful selection of $\sigma$ in our method, some points misclassified initially can also be classified correctly given appropriate $\sigma$ even if the robust radius may not be as large as before. The results on MNIST can be found in Appendix B.1.

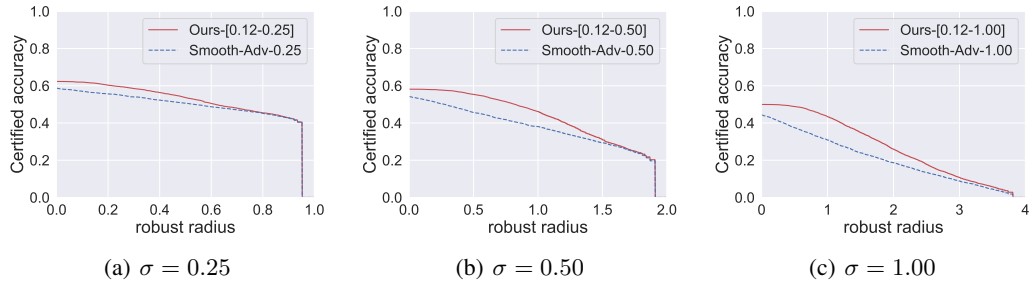

(a) $\sigma = 0.25$        (b) $\sigma = 0.50$        (c) $\sigma = 1.00$

Figure 2: Comparing our sample-wise method with Smooth-Adv on CIFAR-10.

Table 1 stands for the highest certified accuracy under a given $l_2$ perturbation with all hyper-parameters. On CIFAR-10, our sample-wise model outperforms Smooth-Adv significantly, especially with large perturbations. However, on MNIST, our sample-wise model performs worse with large perturbations. We suspect that this may be due to the following reasons: first MNIST dataset has a lower dimensionality and hence is relatively easy, and existing methods already achieve fairly good robustness. But the distance between each two data points in MNIST is smaller that in our method we have to reduce large robust radius to ensure there is no intersection between different regions (see in Eq. (6) and Eq. (8)). So there is a drop on MNIST and the details can be found in Appendix C.2.

| Dataset | Method | 0.00 | 0.25 | 0.50 | 0.75 | 1.00 | 1.25 | 1.50 | 1.75 | 2.00 | 2.25 |
|---|---|---|---|---|---|---|---|---|---|---|---|
| CIFAR-10 | Smooth-Adv(%) | **87** | 73 | 58 | 48 | 38 | 33 | 29 | 24 | 18 | 16 |
| | Ours(%) | 84 | **74** | **61** | **52** | **45** | **41** | **36** | **32** | **27** | **23** |
| MNIST | Smooth-Adv(%) | 99 | 99 | 98 | **96** | **93** | **88** | **81** | **66** | **45** | **37** |
| | Ours(%) | 99 | 99 | 98 | 94 | 88 | 80 | 70 | 56 | 38 | 23 |

Table 1: Upper envelopes of certified accuracy over all experiments on CIFAR-10 and MNIST. The certified accuracy is calculated by sampling 100,000 noise samples for each test set sample.

Table 2 and 3 show the comparison on the ACR metric with single model. The results of Macer on two datasets and results of Smooth-Adv on CIFAR-10 are directly quoted from (Zhai et al., 2020). We implement Smooth-Adv on MNIST with LeNet and report the results as shown in Table 3. The details of models can be found in Appendix A.

| $\sigma$ | Method | 0.00 | 0.25 | 0.50 | 0.75 | 1.00 | 1.25 | 1.50 | 1.75 | 2.00 | 2.25 | ACR |
|---|---|---|---|---|---|---|---|---|---|---|---|---|
| 0.25 | Smooth-Adv(%) | 74 | 67 | 57 | 47 | 0 | 0 | 0 | 0 | 0 | 0 | 0.538 |
| 0.25 | Macer(%) | 81 | 71 | 59 | 43 | 0 | 0 | 0 | 0 | 0 | 0 | 0.556 |
| 0.12-0.25 | Ours(%) | 84 | 74 | 58 | 40 | 0 | 0 | 0 | 0 | 0 | 0 | **0.557** |
| 0.50 | Smooth-Adv(%) | 50 | 46 | 44 | 40 | 38 | 33 | 29 | 23 | 0 | 0 | 0.709 |
| 0.50 | Macer(%) | 66 | 60 | 53 | 46 | 38 | 29 | 19 | 12 | 0 | 0 | 0.726 |
| 0.12-0.50 | Ours(%) | 71 | 69 | 61 | 52 | 42 | 33 | 24 | 17 | 0 | 0 | **0.840** |
| 1.00 | Smooth-Adv(%) | 45 | 41 | 38 | 35 | 32 | 28 | 25 | 22 | 19 | 17 | 0.787 |
| 1.00 | Macer(%) | 45 | 41 | 38 | 35 | 32 | 29 | 25 | 22 | 18 | 16 | 0.792 |
| 0.12-1.00 | Ours(%) | 52 | 52 | 51 | 49 | 45 | 41 | 36 | 31 | 26 | 22 | **1.111** |

Table 2: Certified accuracy and average certified radius(ACR) on CIFAR-10. Every row is a single model. The results of Smooth-adv and Macer are quoted from (Zhai et al., 2020).

| $\sigma$ | Method | 0.00 | 0.25 | 0.50 | 0.75 | 1.00 | 1.25 | 1.50 | 1.75 | 2.00 | 2.25 | ACR |
|---|---|---|---|---|---|---|---|---|---|---|---|---|
| 0.25 | Smooth-Adv(%) | 99 | 99 | 98 | 96 | 0 | 0 | 0 | 0 | 0 | 0 | **0.929** |
| 0.25 | Macer(%) | 99 | 99 | 97 | 95 | 0 | 0 | 0 | 0 | 0 | 0 | 0.918 |
| 0.12-0.25 | Ours(%) | 99 | 99 | 98 | 95 | 0 | 0 | 0 | 0 | 0 | 0 | 0.922 |
| 0.50 | Smooth-Adv(%) | 99 | 98 | 97 | 96 | 93 | 88 | 81 | 66 | 0 | 0 | **1.688** |
| 0.50 | Macer(%) | 99 | 98 | 96 | 94 | 90 | 83 | 73 | 50 | 0 | 0 | 1.583 |
| 0.12-0.50 | Ours(%) | 99 | 99 | 97 | 94 | 87 | 79 | 69 | 52 | 0 | 0 | 1.581 |
| 1.00 | Smooth-Adv(%) | 93 | 91 | 87 | 82 | 76 | 69 | 51 | 53 | 45 | 37 | 1.792 |
| 1.00 | Macer(%) | 89 | 85 | 79 | 75 | 69 | 61 | 54 | 45 | 36 | 28 | 1.520 |
| 0.12-1.00 | Ours(%) | 98 | 97 | 96 | 93 | 88 | 81 | 73 | 57 | 41 | 25 | **1.810** |

Table 3: Certified accuracy and average certified radius(ACR) on MNIST. Every row is a single model. The result of Macer is quoted from (Zhai et al., 2020).

On CIFAR-10, as shown in Table 2, sample-wise model performs better than others on ACR. With lower noise levels $\sigma \in \{0.25, 0.50\}$, it shows the accuracy-robust trade-off that the higher the accuracy of the model, the relative robustness under larger disturbance will be weakened. However with large $\sigma = 1.00$, sample-wise method makes overall progress and promotes ACR with a great margin 0.319, nearly 40% improvement. We argue that with a higher noise level, sample-wise method has more choices for noise selection to provide a better accuracy-robustness trade-off.

On MNIST, as shown in Table 3, sample wise method performs nearly the same as Macer with $\sigma \in \{0.25, 0.50\}$ though worse than Smooth-Adv. The sample-wise method can still achieve better results with high noise level $\sigma = 1.00$ meaning more selection choices.

## 6 CONCLUSION

In this work, we propose sample-wise randomized smoothing technique which selects different noise level for every sample during both training and testing. By allocation of robust regions and assignment of $\sigma$, our sample-wise model achieves better accuracy-robustness trade-off especially with high noise level. Through comparisons, our sample-wise model promotes ACR with a great margin, nearly 40% improvement on CIFAR-10, and achieves comparable results on MNIST.

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

## A    APPENDIX OF EXPERIMENTAL DETAILS

### A.1    EXPERIMENT SETTINGS

Following the experiment settings in (Salman et al., 2019; Zhai et al., 2020), we use the same base classifier ResNet-110 for CIFAR-10 and LeNet for MNIST. Based on Smooth-Adv, we use the project gradient descend(PGD) as the inside attack algorithm. And we have conducted lots of experiments with a set of hyper-parameters. For the attacking steps $k$ of PGD, we have conducted with $k \in \{2, 4, 10\}$. For the maximum $l_2$ perturbation denoted $\epsilon$, we set it with $\epsilon \in \{0.25, 1.0, 2.0\}$. As for the number of noise samples $m$ in Smooth-Adv, we set it with $m \in \{2, 4\}$.

As for the standard deviation $\sigma$, we modify the fixed standard deviation $\sigma$ to a range $[\sigma_{min}, \sigma_{max}]$. And we have conducted with $(\sigma_{min}, \sigma_{max}) \in \{(0.12, 0.25), (0.12, 0.50), (0.12, 1.00)\}$.

As for the pretrain-to-finetune framework, we first pretrain the base model with random $\sigma$ with 150 epochs and finetune the model with specific $\sigma$ with another 150 epochs. And we set the initial learning rate 0.1 for pretrain stage and 0.001 for finetune stage. For both stages, the learning rate drops by a factor of 10 every 50 epochs which is the same as that in (Salman et al., 2019).

For certification, we use 500 noise samples for training stage to assign specific $\sigma$ for every train samples. As for testing stage, actually, we first certify the test set sample $x^j$ with varying $\sigma \in [\sigma_{min}, \sigma_{max}]$ with only 1000 noise samples. After selecting the best $\sigma^j$, we do another certification procedure with 100,000 noise samples to get robust radius. As for the allocation order, we apply the ascending order defined in Appendix C.2.

We use the same hyper-parameters for both CIFAR-10 and MNIST. And we implement Smooth-Adv on MNIST with the same settings, except that $\sigma \in \{0.12, 0.25, 0.50, 1.00\}$.

### A.2    COMPARING MODELS

In this section, we list the models that are used in the comparison of average certified radius(ACR). Table 4 shows the exact models that are used for CIFAR-10. The results of Smooth-Adv and Macer are directly quoted from Zhai et al. (2020). Table 5 shows the exact models that are used for MNIST. The result of Macer is quoted from Zhai et al. (2020), and the result of Smooth-Adv is got by the implementation of ourselves.

| $\sigma$ | Method | Description |
|---|---|---|
| 0.25 | Smooth-Adv | 8-sample 10-step Smooth-Adv with $\epsilon = 1.00$ |
| 0.25 | Macer | Macer with $k = 16$, $\lambda = 12.0$, $\beta = 16.0$ and $\gamma = 8.0$ |
| 0.12-0.25 | Ours | 4-sample 4-step sample-wise model with $\epsilon = 0.25$ |
| 0.50 | Smooth-Adv | 2-sample 10-step Smooth-Adv with $\epsilon = 2.00$ |
| 0.50 | Macer | Macer with $k = 16$, $\lambda = 4.0$, $\beta = 16.0$ and $\gamma = 8.0$ |
| 0.12-0.50 | Ours | 4-sample 2-step sample-wise model with $\epsilon = 1.00$ |
| 1.00 | Smooth-Adv | 2-sample 10-step Smooth-Adv with $\epsilon = 2.00$ |
| 1.00 | Macer | Macer with $k = 16$, dynamic $\lambda$, $\beta = 16.0$ and $\gamma = 8.0$ |
| 0.12-1.00 | Ours | 2-sample 4-step sample-wise model with $\epsilon = 2.00$ |

Table 4: Compared models on CIFAR-10 for ACR comparison.

## B    APPENDIX OF EXPERIMENTS RESULTS

In this section, we show more results of the comparison with Smooth-Adv on MNIST. And through several ablation experiments, we study the effects of different parts in the provable robustness.

### B.1    COMPARISON WITH SMOOTH-ADV

We compare our sample-wise method with Smooth-Adv given the same hyper-parameters except $\sigma$. The models shown in Fig. 3 are both trained with $k = 10$, $m = 2$ and $\epsilon = 2.0$. The Smooth-

| $\sigma$ | Method | Description |
|---|---|---|
| 0.25 | Smooth-Adv | 4-sample 10-step Smooth-Adv with $\epsilon = 0.25$ |
| 0.25 | Macer | Macer with $k = 16$, $\lambda = 16.0$, $\beta = 16.0$ and $\gamma = 8.0$ |
| 0.12-0.25 | Ours | 4-sample 2-step sample-wise model with $\epsilon = 0.25$ |
| 0.50 | Smooth-Adv | 4-sample 10-step Smooth-Adv with $\epsilon = 1.00$ |
| 0.50 | Macer | Macer with $k = 16$, $\lambda = 16.0$, $\beta = 16.0$ and $\gamma = 8.0$ |
| 0.12-0.50 | Ours | 4-sample 2-step sample-wise model with $\epsilon = 1.00$ |
| 1.00 | Smooth-Adv | 4-sample 10-step Smooth-Adv with $\epsilon = 2.00$ |
| 1.00 | Macer | Macer with $k = 16$, $\lambda = 16.0$, $\beta = 16.0$ and $\gamma = 8.0$ |
| 0.12-1.00 | Ours | 4-sample 10-step sample-wise model with $\epsilon = 2.00$ |

Table 5: Compared models on MNIST for ACR comparison.

Adv models are trained with $\sigma \in \{0.25, 0.50, 1.00\}$. Correspondingly, we train our model with $(\sigma_{min}, \sigma_{max}) \in \{(0.12, 0.25), (0.12, 0.50), (0.12, 1.00)\}$. With small $\sigma = 0.25$, these two methods perform nearly the same. With $\sigma = 0.50$, Smooth-Adv performs better with promotion on large perturbation. With $\sigma = 1.00$, our method performs better with small perturbation and Smooth-Adv performs better with large perturbation. The reason why it is worse on large perturbation on MNIST is that the data points in MNIST are simpler and it is easier to obtain a higher robust radius. However in our algorithm, in order to make the regions have no intersection, the larger robust radius will be sacrificed.

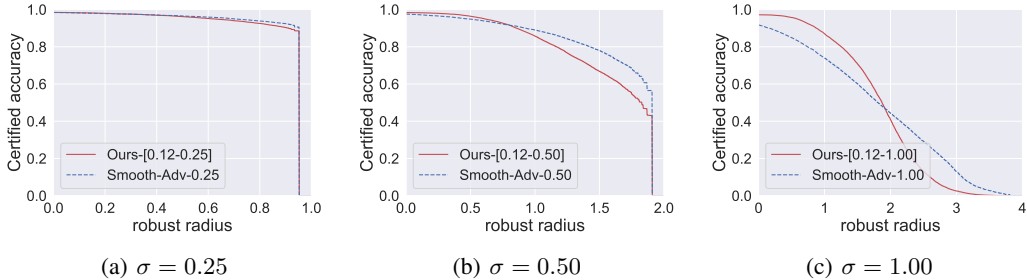

(a) $\sigma = 0.25$      (b) $\sigma = 0.50$      (c) $\sigma = 1.00$

Figure 3: Comparing our sample-wise method with Smooth-Adv(Salman et al., 2019)

## B.2 ABLATION STUDY

As aforementioned, our method consists of two parts: the first one is pertrain-to-finetune framework and the second one is sample-wise certification procedure. Thus, we study the effects of different parts on CIFAR-10 and MNIST as shown in Fig. 4 and Fig. 5. We have conducted the following four algorithms:

**Ours-diff**: Our pretrain-to-finetune model test under sample-wise certification procedure, with specific different noise level for each test data.

**Ours-same**: Our pretrain-to-finetune model test with the same noise level for all test data (conventional certification procedure).

**Smooth-Adv-diff**: The Smooth-Adv model test under our sample-wise certification procedure.

**Smooth-Adv-same**: The Smooth-Adv model test with the same noise level for all test data.

From the results on CIFAR-10 shown in Fig. 4, our sample-wise certification procedure can still slightly promote the performance of Smooth-Adv and our pretrain-to-finetune model test under the same noise level performs nearly the same as Smooth-Adv. From these, the promotion of individual part is limited. Only by combining the pretrain-to-finetune framework with sample-wise certification procedure can our model achieve a larger improvement.

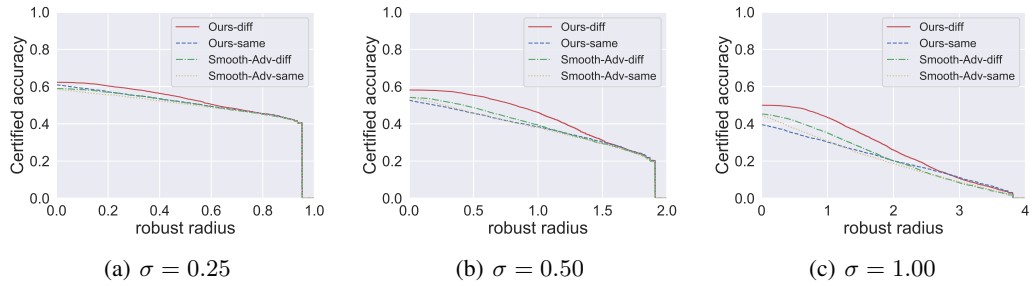

(a) $\sigma = 0.25$        (b) $\sigma = 0.50$        (c) $\sigma = 1.00$

Figure 4: The effects of different components in our model on CIFAR-10.

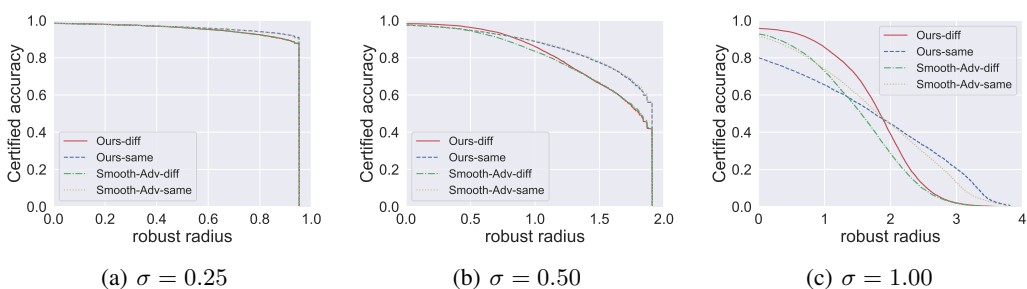

(a) $\sigma = 0.25$        (b) $\sigma = 0.50$        (c) $\sigma = 1.00$

Figure 5: The effects of different components in our model on MNIST.

Fig. 5 shows the comparison on MNIST. With lower noise level, there is nearly no difference between sample-wise certification and conventional certification. With high noise level $\sigma = 1.00$, sample-wise certification weaken the robustness of Smooth-Adv but significantly improves the performance of our model, which also show that the two parts should be applied at the same time.

## C  EFFECT OF ORDER

Since our algorithm needs to generate regions for each test sample one by one, the order of allocation will also affect the performance of the provable robustness. In this section, we use several experiments to illustrate the effect of the order and elaborate the distance from the optimal results.

### C.1  PROBLEM REDEFINITION

Suppose that we do not consider whether each sample in test set is correctly classified, and we do not consider the standard deviation assigned to each sample point. The main purpose is to allocate non-overlapping regions around each sample and maximum the sum of the radii of these regions. Described in mathematical language as follows:

The test set is denoted as $\{x^j\}_{1 \le j \le n}$. Define $D(x, y)$ as the Euclidean distance between two points $x$ and $y$. Allocate specific neighborhood $B(x^j, r^j) = \{x : D(x, x^j) < r^j\}$. The radius $\{r^j\}$ is exactly the variable and it satisfies that $0 \le r^j \le R^j (1 \le j \le n)$. Then the optimization objective is as shown in (10)

$$\text{maximize } \frac{1}{n} \sum_{j=1}^{n} r^j$$
$$\text{s.t.} \quad B(x^i, r^i) \cap B(x^j, r^j) = \emptyset, \qquad \forall i \ne j \tag{10}$$

### C.2  RESULTS OF DIFFERENT ORDER

Following the definition in previous section, we study the effects of the different sequences order. The variable $R$ is exactly the maximum robust radius calculated with $\sigma \in [\sigma_{min}, \sigma_{max}]$. And we transform the optimization problem to an undirected graph $G(V, E)$, exactly $G_{CIFAR-10}$ and

$G_{MNIST}$. If $B(x^i, R^i)$ has intersection with $B(x^j, R^j)$, then we add $x^i$ and $x^j$ to $V$ and add the edge $\overline{x^i x^j}$ to $E$. Then we can plot the distribution of the degree in $G$ for CIFAR-10 and MNIST as shown in Figure 6. And the models we use are the same as those in Table 4 and Table 5 with $\sigma$ ranges from 0.12 to 1.00.

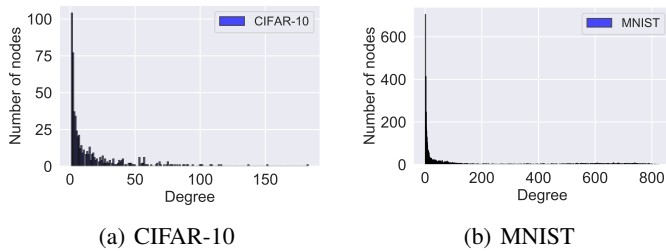

(a) CIFAR-10         (b) MNIST

Figure 6: Distribution of the degree in $G$ for CIFAR-10 and MNIST

Figure 6 shows that most of the graph nodes have a low degree and only a few nodes have a very large degree, especially the degree even exceeds 800 in MNIST. Exactly, there are 569 nodes in $G_{CIFAR-10}$ and 4777 nodes in $G_{MNIST}$ which means that only $5\%$ regions intersect others in CIFAR-10 but nearly $47\%$ regions intersect others in MNIST if without post-processing. Due to the high overlap between different regions in MNIST, our sample-wise method performs poorly on MNIST.

Actually, the optimization function in Eq. 10 is a standard linear programming problem. We can get the theoretical optimal solution by solving the linear programming problem. However, especially for MNIST, there exits nearly 4700 variables and more than 300,000 limits, making this problem difficult. As it is hard to optimize such many variables in the same time, we switch to a online setting which optimizes variables one by one. The online setting is more practical and is just what we have described in the main body. The online optimization objective is briefly described as Eq. (11):

$$
\begin{aligned}
& maximize\ r^k\ (1 \le k \le n) \\
s.t.\ & \\
& 0 \le r^k \le R^k \\
& B(x^i, r^i) \cap B(x^k, r^k) = \emptyset\ (1 \le i < k)
\end{aligned}
\tag{11}
$$

Based on the degree of graph nodes, we use the following two special orders to estimate the results.

1. (Descending order) Degree in descending order.

2. (Ascending order) Degree in ascending order.

And the test results is shown in Table 6. The type of **Overlapping** means that the $r^j = R^j$ regardless of the overlapping of regions. **Descending** represents the order that test samples come with descending degree and **Ascending** represents the reverse order. The results shows that the regional overlap affects the performance a lot. On CIFAR-10, the difference between two orders is small as there is a few intersections. On MNIST, the ascending order performs a little bit better. From the results, we choose to apply the ascending order into our algorithm in the main body.

| Dataset | Types | 0.00 | 0.25 | 0.50 | 0.75 | 1.00 | 1.25 | 1.50 | 1.75 | 2.00 | 2.25 | ACR |
|---|---|---|---|---|---|---|---|---|---|---|---|---|
| | Overlapping | 100 | 99.5 | 96.2 | 89.4 | 79.9 | 69.6 | 59.7 | 50.4 | 42.1 | 35.0 | 1.933 |
| CIFAR-10 | Descending | 94.4 | 93.9 | 90.6 | 83.8 | 74.3 | 64.0 | 54.1 | 44.8 | 36.5 | 29.5 | 1.737 |
| | Ascending | 94.3 | 93.8 | 90.6 | 83.7 | 74.3 | 63.9 | 54.0 | 44.7 | 36.5 | 29.4 | 1.734 |
| | Overlapping | 100 | 100 | 100 | 100 | 100 | 100 | 99.7 | 98.4 | 92.6 | 82.6 | 2.811 |
| MNIST | Descending | 97.4 | 96.2 | 94.0 | 90.0 | 83.7 | 77.1 | 69.6 | 59.6 | 45.7 | 30.8 | 1.821 |
| | Ascending | 99.9 | 99.5 | 97.9 | 93.3 | 86.4 | 79.6 | 71.9 | 63.4 | 50.3 | 35.1 | 1.917 |

Table 6: Certified accuracy and ACR on CIFAR-10 and MNIST with different orders.

# D    DISCUSSION OF TRANSDUCTIVE SETTING

The transductive setting means that we know the test data points except their labels. In this setting, we explore which noise level could be more suitable for each sample. As we discussed in detail before, we divide the input space into "robust regions" according to the properties of test data. Hence, in this setting, we can solve a linear program based on the given test data (See Appendix C).

In the setting that test data points come online, we give up global optimization (i.e., solving the linear program for all test data points). In this case, we have to allocate regions for test data one by one. As we discussed in Section 4.2, the classification of a testing point depends on which "robust regions" it lies in. Hence, one can imagine that the robustness of a testing point is limited by the distance between it and the boundary of its allocated region. As the number of test data increases, those closer to some region boundary would get a smaller radius and it is plausible that the average robust radius could gradually decrease. Hence, this may be seen as a potential weakness of our sample-wise algorithm.

Nevertheless, we explore how much the size of the test set affects our performance in the following experiments. We extend CIFAR-10 with 20,000 images, whose categories are subset of CIFAR-10, from ImageNet. We directly use our trained sample-wise model to certify all these 30,000 images (10,000 from CIFAR-10 and 20,000 from ImageNet) using linear programming or with the online setting. The results are shown in Table 7.

| Dataset | Types | 0.00 | 0.25 | 0.50 | 0.75 | 1.00 | 1.25 | 1.50 | 1.75 | 2.00 | 2.25 |
|---|---|---|---|---|---|---|---|---|---|---|---|
| CIFAR-10-Extra | Linear Programming | 41.03 | 40.78 | 39.74 | 37.43 | 34.03 | 30.10 | 26.04 | 22.18 | 18.68 | 15.54 |
| | Online setting | 40.98 | 40.80 | 39.76 | 37.44 | 34.04 | 30.06 | 26.00 | 22.12 | 18.58 | 15.41 |

Table 7: Certified accuracy on extension dataset with linear programming or online setting.

Table 7 shows that the performance with online setting and linear programming are nearly the same on a test of 30,000 images. This also shows that even in the online setting, the impact of increasing the test data set on performance is small, and the performance degradation is slow. In particular, with 30,000 test points, we transform the result to a graph as described in Appendix C and there are only 1388 data points intersect others, affecting less than $5\%$ of all testing points.

To sum up, our sample-wise algorithm works in the transductive setting and the setting where the test data points come online if there are a reasonable number of testing data points. In the extreme case that there are drastically more test points, it is possible to further leverage these unlabeled data to enhance the robustness via semi-supervised learning (Carmon et al., 2019; Zhai et al., 2019), and we leave it as an interesting future direction.

