# OpenReview forum: "Pretrain-to-Finetune Adversarial Training via Sample-wise Randomized Smoothing"
_ICLR.cc/2021/Conference — Reject_

### Official Review · AnonReviewer3 · 2020-10-27
**Motivation and explanations of methodology could be improved**

**Rating:** 4
**Confidence:** 4

**Review:**

This paper proposes an improved sample-wise randomized smoothing technique, where the noise level is tuned for different samples, for certification of robustness. Further, it also proposes a pretrain-to-finetune methodology for training networks which are then certified via sample-wise randomized smoothing. The authors show in experiments on CIFAR and MNIST that combining their training methodology and certification methodology can sometimes improve the average certified when compared to state-of-the-art randomized smoothing techniques Smooth-Adv (Salman et. al, 2019).

I recommend a rejection because the key takeaways of the paper should be clarified and the pretrain-to-finetune framework and the allocation of regions must be explained and justified better.

The key idea of using different noise levels for different samples is intuitive and explained well in the motivation section (4.1). Furthermore, the authors show that their methodology does indeed lead to minor improvements in average certified l2-radius on Smooth-Adv for the CIFAR dataset, which is a more interesting dataset than MNIST, where the proposed technique performs similarly or slightly worse than Smooth-Adv.

However, the paper does have shortcomings in its clarity and organization. First, I think the sample-wise certification is a clear and well-motivated idea, and should be discussed as the major contribution, rather than the pretrain-to-finetune framework. Furthermore, I was confused about the allocation of regions in the prediction step of the sample-wise certification; explaining why it is necessary, and why it is better than allocating a region for every single test datapoint (which is what I thought the motivation section in 4.1 explained) would improve the paper significantly. Finally, the amount of notation in the paper should be simplified significantly, and the notation often makes the paper more confusing (and sometimes, I could not understand due to either incorrect or unclear notation). For example, the pseudocode in Algorithm 1 would have been better if the notation was simplified, and in Algorithm 2, I did not know what B_{i_j} referred to at all.

Specifically regarding the allocation of regions, I did not understand why it was necessary or led to improvements over choosing a new region for each test datapoint. Explaining it clearly, and showing an ablation study that compares using region-allocation and not using region-allocation would provide good motivation for its use.

Specifically regarding the pretrain-to-finetune framework, I have the following questions:
I saw that in Appendix C that the pretrain-to-finetune framework is necessary for the sample-wise randomized smoothing to show an improvement. Are there explanations for why sample-wise randomized smoothing does not well work by itself?

Why does it make sense to do this 2 step procedure? Why does the pre-training have to involve varying noise levels if the fine-tuning procedure already finds the optimal noise level for each sample to train with? Could the pre-training just be the same as Smooth-Adv?

How much does it matter which noise levels we choose during the pre-training phase? I noticed that the authors usually chose noise from 0.12 up to the amount that they compare to with SmoothAdv, but the reasons for this are not discussed.


Overall, I feel that the paper has a well-motivated idea (sample-wise randomized smoothing) and shows some minor improvements in terms of results, but that clarity for all other parts of the paper must be improved significantly.


Post Rebuttal Update:

I appreciate the author response, but I will maintain my score after reading the rebuttal and discussion with other reviewers. It still appears to me that the motivation and clarity can be improved, and so I would recommend focusing on those aspects in future revisions. Additionally, baselines such as "allocating a region for every single test point" should be compared to in a clear way (as opposed to being in the appendix), as such baselines seem natural to compare to.

---

> ### Author Response · Authors · 2020-11-24
> **Response to AnonReviewer3**
>
> Thanks for your detailed and insightful comments. Here are the responses to your concerns.
>
> 1. **Regions Allocation**. We first explain the necessity of the allocation of regions in the prediction step. According to the proof of the randomized smoothing theorem in (Cohen et al., 2019)[1], we cannot assign an arbitrary noise level to any test point; a certain robustness radius around a point can be certified only if all points within the radius are assigned the same Gaussian variance. Otherwise, it is not certifiable. So, we allocate regions in prediction to ensure correctness.
> 2. Allocating a region for every single test point is also fine, but these regions would be smaller to ensure there is no intersection between different regions.
> 3. **$B_{i_j}$**. $B_{i_j}$ stands for the robust region for the test point $x_{test}^j$. It belongs to a larger region $B_i$. All points in the region $B_i$ use the same sigma.
> 4. **Necessity of Pretraining**. For the pretrain-to-finetune framework, if we remove the pretrain-to-finetune framework which in fact is the Smooth-Adv-diff which is shown in appendix B.2,  it performs nearly the same as the original Smooth-Adv. We think that Smooth-Adv-diff is trained with a fixed sigma, even if applying the sample-wise method in prediction, the model still prefers the sigma used in training which makes the sample-wise ineffective.
> 5. **Varying Sigma**. Since different standard deviations are used in the testing process, we use the varying sigma in training expecting that the model can find the best sigma for different points. The finetune procedure is to select the best sigma based on the results of the pretrain process, hence, if we do not pretrain a base model, we cannot even select sigmas. About using Smooth-adv as the pretrain model, we think that the finetune model may prefer the sigma used in pretraining which may limit the performance of the model.
> 6. **Noise Level**. The reason that we always choose sigma 0.12 as the starting point is that Smooth-Adv uses 0.12 as their smallest noise level. In order to maintain the consistency of the noise range, we directly use 0.12 as our minimum noise.
>
> References:
>
> [1] Jeremy M Cohen, Elan Rosenfeld, and J Zico Kolter. Certified adversarial robustness via randomized smoothing. arXiv preprint arXiv:1902.02918, 2019

---

### Official Review · AnonReviewer1 · 2020-10-27
**Interesting work but requires more clarifications**

**Rating:** 6
**Confidence:** 4

**Review:**

This paper considers the problem of provably defense to adversarial perturbations using randomized smoothing. The authors propose sample-wise randomized smoothing -- assigning different noise levels to different samples. They also propose to first pretrain a model and then adjust the noise for higher performance based on the model’s outputs. Experiments show that proposed approach improves the performance of randomized smoothing with same noise level for small perturbations.

Pros:
1)	The paper is well written and easy to read.
2)	The idea of sample-wise randomized smoothing is interesting, and results are reasonable.
3)	Issues with assigning arbitrary noise level to test points is well described/thought and solutions (online and batchwise) are proposed to make it compatible.
4)	Experimental setup is comprehensive and appropriate ablation studies have been performed.

Cons:
1)	My main concern with this work is that it is not clear to me that these ACR gains are being achieved at what cost? It appears that sample-wise randomized smoothing adds an additional computational complexity during both training and prediction/certification phases. I would like to see the train and prediction cost comparison with standard train/test, MACER, and vanilla random smooth model. This comparison will provide a better insight into the performance as on some cases, e.g., MNIST, the sample-wise RS performs pretty close to the baselines. I will argue that in the computation cost on sample-wise RS is significantly higher than the baseline robust approaches, one can simply increase the m_test in those approaches.
2)	It will insightful to see how much gain the proposed scheme achieves with vanilla gaussian augmented models (authors only show these results with smooth adversarially trained models).
3)	Similar to adv-smooth, it will be useful to see how much gain can be achieved with: 1) pre-training, and 2) semi-supervised learning.
4)	Results in Sec 5.2 is for online or batch setting?
5)	How does resolution of grid or \sigma_interval impact the performance (train and prediction/certification time and ACR/ACA)?

Minor:
1)	There seems to be typo in Sec 5.1: [0, 12, 0.25].

---

> ### Author Response · Authors · 2020-11-24
> **Response to AnonReviewer1**
>
> Thanks for your detailed and insightful comments. Here are the responses to your concerns.
>
> 1. **Computation Cost**. In our method, the computation cost mainly comes certification procedure both in training and prediction. Take CIFAR-10 for example. In training, we have to certify every train set datapoint, which is 5 times the test set size. We choose 0.05 as the sigma interval which means we have to certify 20 different sigmas and we only use 500 samples which are 1/200 of 10w samples. So, it costs nearly $5 * 20 * 1/200 = 0.5$ times comparing with certification procedure on test set proposed by (Cohen et al., 2019)[1]. In prediction, we can also first use 500 samples to assign sigma and then use 10w samples in certification, so it costs nearly 1.1 times the time of certification. To sum up, taking the pretrain-to-finetune framework into account, it is nearly 2 times the computation cost compared with Smooth-Adv.
>
> 2. **Implementation on Vanilla Gaussian Augmented Models**. We implemented our framework based on the (Cohen et al., 2019)[1]. For CIFAR-10 and MNIST, we use the maximum noise level 1.00, and the results are shown below:
>
>    | Dataset  | Model | 0    | 0.25 | 0.50 | 0.75 | 1.0  | 1.25 | 1.5  | 1.75 | 2.0  | 2.25 | ACR       |
>    | -------- | ----- | ---- | ---- | ---- | ---- | ---- | ---- | ---- | ---- | ---- | ---- | --------- |
>    | MNIST    | Cohen | 0.95 | 0.92 | 0.87 | 0.81 | 0.72 | 0.61 | 0.50 | 0.34 | 0.20 | 0.10 | 1.417     |
>    | MNIST    | Ours  | 0.99 | 0.99 | 0.97 | 0.93 | 0.85 | 0.74 | 0.60 | 0.42 | 0.25 | 0.12 | **1.609** |
>    | CIFAR-10 | Cohen | 0.47 | 0.39 | 0.34 | 0.28 | 0.21 | 0.17 | 0.14 | 0.08 | 0.05 | 0.03 | 0.458     |
>    | CIFAR-10 | Ours  | 0.70 | 0.67 | 0.58 | 0.48 | 0.38 | 0.30 | 0.24 | 0.18 | 0.14 | 0.10 | **0.900** |
>
>    From the results, our method achieves a significant improvenent over ACR. In particular, it outperforms (Cohen et al., 2019)[1] significantly on CIFAR-10 with small $l_2$ radius. For ACR on CIFAR-10, our sample-wise method based on (Cohen et al., 2019)[1] outperforms Smooth-Adv but is still worse than our sample-wise method based on Smooth-Adv.
>
> 3. According to the results on CIFAR-10 reported in (Salman et al., 2019)[2] (as follow), our sample-wise method outperforms the pre-training and semi-supervised methods when the $l_2$ radius is larger than 0.75. It would be interesting to see how much gains can be achieved with these two methods especially with smaller $l_2$ radius and we leave it as an interesting future direction.
>
>    | Model              | 0.25   | 0.5    | 0.75   | 1.0    | 1.25   | 1.5    | 1.75   | 2.0    | 2.25   |
>    | ------------------ | ------ | ------ | ------ | ------ | ------ | ------ | ------ | ------ | ------ |
>    | SmoothAdv          | 73     | 58     | 48     | 38     | 33     | 29     | 24     | 18     | 16     |
>    | + Pre-Training     | 80     | 62     | **52** | 38     | 34     | 30     | 25     | 19     | 16     |
>    | + Semi-supervision | 80     | **63** | **52** | 40     | 34     | 29     | 25     | 19     | 17     |
>    | + Both             | **81** | **63** | **52** | 37     | 33     | 29     | 25     | 18     | 16     |
>    | Ours               | 74     | 61     | **52** | **45** | **41** | **36** | **32** | **27** | **23** |
>
> 4. **Settings**. The results showing in Sec 5.2 are obtained in the online setting.
>
> 5. **Sigma Interval**. Sigma interval mainly controls the computation cost. If we halve the interval, it takes double time to allocate the standard deviation. As for ACR/ACA, we suspect that they would be improved as we can choose sigma more accurately.
>
> References:
>
> [1] Jeremy M Cohen, Elan Rosenfeld, and J Zico Kolter. Certified adversarial robustness via randomized smoothing. arXiv preprint arXiv:1902.02918, 2019
>
> [2] Hadi Salman, Jerry Li, Ilya Razenshteyn, Pengchuan Zhang, Huan Zhang, Sebastien Bubeck, and Greg Yang. Provably robust deep learning via adversarially trained smoothed classifiers. In Advances in Neural Information Processing Systems, pp. 11289–11300, 2019.

---

### Official Review · AnonReviewer4 · 2020-11-03
**Nice but straightforward extension of existing method**

**Rating:** 5
**Confidence:** 2

**Review:**

The paper propose a method to improve the randomized smoothing algorithm for certified robustness against adversarial attacks.
The idea is that, instead of adding the same Gaussian noise to every data points, it uses a different standard deviation for each data points. When an example is far away from the decision boundary, one can add more noise.
Pros:
- Certified robustness is an important problem in adversarial ML, and randomized smoothing is one of most promising methods.
- The proposed method is intuitive and seems to be a practical way to improve the original randomized smoothing algorithm
- Experiments show that, the certified accuracy on CIFAR-10 really increases
Cons:
- It seems to me that the proposed method is a relatively straightforward extension from the original randomized smoothing algorithm, so the technical contribution is limited.

---

> ### Author Response · Authors · 2020-11-24
> **Response to AnonReviewer4**
>
> Thanks for your detailed and insightful comments. Here are the responses to your concerns.
>
> 1. In this work, our sample-wise method is applied to the current optimal model Smooth-Adv, but the method we propose does not depend on Smooth-Adv. We believe that our method can be used as a tool and directly applied to other methods to improve certifiable robustness.

---

### Official Review · AnonReviewer5 · 2020-11-09
**A method for adaptive smoothing parameters in randomized smoothing**

**Rating:** 4
**Confidence:** 4

**Review:**

This paper suggests an extension of randomized smoothing, wherein the degree of smoothing is optimized both at training and test-time on each individual sample. At training time, the model is first "pre-trained" using a range of smoothing parameters (variance of the Gaussian perturbations), and then "fine-tuned" by selecting the variance on each sample which maximizes the verified radius. At test time, we can again select the smoothing parameter to maximize robustness.

Pros:
- Numerically, the results seem fairly strong

Cons:
- It's unclear to me whether the evaluation is fair


A few (somewhat critical) questions:

1. (Major) For the test-time procedure, this procedure selects $\sigma$ based on a computed robustness statistic. I assume that this robustness statistic uses the original image, as in other randomized smoothing approaches? (as opposed to an adversarially perturbed image). If so, this comparison seems somewhat unfair - the typical threat model is that the classifier does not get to first see the nominal image (otherwise, the classifier could cache the clean image + label, and use a nearest-neighbor lookup against its cache to handle any adversarial images.) If not, could you explain how the adversarial image is selected here?

2. What is the purpose of the balls $B$ in the section on "Predicting Procedure."? What do they add compared to computing $r^j$ directly? For what fraction of the test set is an existing $B_i$ found including the test point? (I would expect this fraction to be very small?)

3. It seems that for e.g. the SmoothAdv model trained with $\sigma = 0.25$, we should be interested in robustness with radii significantly below 0.25 (and certainly not above it). Am i misunderstanding the naming of the models?

Minor points:
- It would be interesting to see an ablation of whether the fine-tuning phase helps.
- The presentation of the algorithm could be significantly simplified (lots of notation is unnecessarily complicated, double subscripting, going into details before explaining the idea, lots of new symbols introduced throughout, etc.). The pseudocode is very helpful.

Overall:
It's clear that the authors have put significant effort into this submission, but I believe it does not currently meet the necessary bar for ICLR, though I may adjust my rating if the rebuttal satisfactorily addresses the points above. I hope some of this feedback will be useful to the authors.

EDIT: Thanks for the clarifications. Unfortunately, none of the responses are enough for me to update my rating.
One thing regarding point 1 in particular: the transductive setting seems contrived for adversarial robustness as it does not seem to correspond to a plausible threat model. It's true that in the transductive setting, the examples don't have labels, but since clean accuracy >> robust accuracy, just caching predicted labels on the clean examples is roughly as good (which can be done even if test labels are not available).

---

> ### Author Response · Authors · 2020-11-24
> **Response to AnonReviewer5**
>
> Thanks for your detailed and insightful comments. Here are the responses to your concerns.
>
> 1. **Evaluation**. In the prediction procedure, there are indeed some differences: we present our results in the transductive setting. We assume that the test dataset is known except labels. Hence, we allocate regions and assign different sigmas based on the results of the test dataset. Instead of caching the clean image + label, we only guarantee the points in the same region use the same standard deviation. As for adversarial image, it can be any perturbed image and we can assign a suitable standard deviation for it according to its location.
> 2. **Regions Allocation**. According to the proof of the randomized smoothing theorem in (Cohen et al., 2019)[1], we cannot assign arbitrary noise level to any test point; a certain robustness radius around a point can be certified only if all points within the radius are assigned the same Gaussian variance. So, we use different regions $B$ in the prediction procedure and the datapoints located in the same region are assigned the same standard deviation. In our experiments, the fraction of the test points in an existing $B_i$ is very small.
> 3. **Certified Robust Radius**. As the Smooth-Adv model is trained with $\sigma=0.25$, the distances between most noisy
>    samples and the clean sample are within $3\sigma$. Among these noisy points, if only 50% of the predictions are correct, then the robust radus is significantly small, nearly 0. However, if 99% of the predictions are correct, we can get a larger robust radius(nearly 0.95) according to (Cohen et al., 2019)[1]. So, we do not only focus on the robustness with radii significantly below 0.25.
> 4. **Ablation for Finetuning**.  We have not conducted ablation for finetune procedure as we treat pretrain-to-finetune as a whole, and it is an interesting future direction.
>
> References:
>
> [1] Jeremy M Cohen, Elan Rosenfeld, and J Zico Kolter. Certified adversarial robustness via randomized smoothing. arXiv preprint arXiv:1902.02918, 2019

---

### Comment · ~Mao_Ye12 · 2021-03-13
**Does the robust radius Lemma still holds for the algorithm in this paper?**

Hi Authors,

Thanks for your interesting papers. I have one questions on your paper.

Since the noise level sigma actually depends on the input image x, thus we can view sigma as a function of x. In this case I feel the key robust radius lemma no more holds. The reason is that, given x, for some x' in its l2 balls, the sigma for x' might be different than x and thus result in Cohen's paper seems no more applicable.

Could you clarify that?

Thanks.

---

> ### Comment · ~Lei_Wang22 · 2021-03-14
> **Response to the robust radius theorem.**
>
> Thanks for your question.
>
> Your concern is right and we also mention this issue in the introduction:  a certain robustness radius around a point can be certified only if all points within the radius are assigned the same Gaussian variance.
>
> To address this issue, we divide the input space into 'robust regions' and the samples in the same region are assigned the same noise level. Further, we make sure that the certified l2-ball does not exceed the 'robust region' it falls in so that the theorem still holds in the region.

---

### Decision · Program_Chairs · 2021-01-07
**Final Decision**

**Decision:**

Reject

**Comment:**

The paper considers an extension of randomized smoothing where the smoothing noise may differ for different points. The resulting method shows good performance experimentally. However, the reviewers raised a number of problems which, at the moment, precludes the acceptance of the paper, such as the following:

- The paper analyzes the transductive setting, where all the test points are available to fine-tune the smoothing parameters of the predictor. It is not clear how this setting corresponds to a real adversarial threat model, and whether the final tuning needs to use the perturbed or unperturbed points. In the first case, the resulting certified radius is different from what is normally used in the literature, while in the latter it is not clear how the method would be useful to mitigate any real adversarial attack.
- A related comment is that the paper should explain (and state) properly how the results of Cohen et al.  (2019) are applicable to compute the certified radius, which would also provide a proper explanation why partitioning is used.
- The training cost of the procedure seems very high, and this is not discussed.
- The clarity of the presentation should be improved.